# Effects of Biological Nitrogen Metabolism on Glufosinate-Susceptible and -Resistant Goosegrass (*Eleusine indica* L.)

**DOI:** 10.3390/ijms241813791

**Published:** 2023-09-07

**Authors:** Qiyu Luo, Hao Fu, Fang Hu, Shiguo Li, Qiqi Chen, Shangming Peng, Cunyi Yang, Yaoguang Liu, Yong Chen

**Affiliations:** 1College of Agriculture, South China Agricultural University, Guangzhou 510642, China; qiyuluo@scau.edu.cn (Q.L.); focushower@foxmail.com (H.F.); hfang163@163.com (F.H.); shiguoli@stu.scau.edu.cn (S.L.); ycy@scau.edu.cn (C.Y.); 2College of Life Science, South China Agricultural University, Guangzhou 510642, China; qiqichen@stu.scau.edu.cn (Q.C.); 202015140615@stu.scau.edu.cn (S.P.)

**Keywords:** nitrogen metabolism, herbicide stress, glufosinate resistance, goosegrass

## Abstract

Glufosinate is a broad-spectrum herbicide used to control most weeds in agriculture worldwide. Goosegrass (*Eleusine indica* L.) is one of the top ten malignant weeds across the world, showing high tolerance to glufosinate via different mechanisms that are not yet fully understood. This study revealed that nitrogen metabolism could be a target-resistant site, providing clues to finally clarify the mechanism of glufosinate resistance in resistant goosegrass populations. Compared to susceptible goosegrass (NX), the resistant goosegrass (AUS and CS) regarding the stress of glufosinate showed stronger resistance with lower ammonia contents, higher target enzyme GS (glutamine synthetase) activity, and lower GOGAT (glutamine 2-oxoglutarate aminotransferase) activity. The GDH (glutamate dehydrogenase) activity of another pathway increased, but its gene expression was downregulated in resistant goosegrass (AUS). Analyzing the transcriptome and proteome data of goosegrass under glufosinate stress at 36 h showed that the KEGG pathway of the nitrogen metabolism was enriched in glufosinate-susceptible goosegrass (NX), but not in glufosinate-resistant goosegrass (CS and AUS). Several putative target genes involved in glufosinate stress countermeasures were identified. This study provides specific insights into the nitrogen metabolism of resistant goosegrass, and gives a basis for future functional verification of glufosinate-tolerance genes in plants.

## 1. Introduction

The production and security of global crops are seriously affected by abiotic stresses, including herbicide stress [1]. Meanwhile, production costs have increased with the evolution of resistant weeds, caused by the extensive use of herbicides [2,3]. Globally, herbicide-resistant weeds have developed resistance to 21 out of 31 known herbicide action sites from 165 different herbicides, covering 96 crops in 71 countries [4]. The central components of the plant trigger the xenome, xenobiotic detection, transport, and detoxification network, responding to abiotic stresses such as herbicides [3,5,6]. The mechanisms of herbicide resistance in weeds are usually divided into target site resistance and non-target site resistance [7]. To solve these problems, there are some possibilities for crop improvement that can continue being explored alongside more herbicide-resistant genes, allowing to understand in depth the mechanisms of evolved herbicide resistance in weeds.

Although glufosinate has been used for over 40 years, glufosinate remains a key and widely used non-selective herbicide to manage noxious and glyphosate-resistant weeds in agricultural or non-agricultural systems [8,9]. Since glufosinate was first commercialized in the United States and Canada in 1993–1994, it has played a critical role in tropical countries where grasses are the biggest challenge for weed control; paraquat in particular was widely used, although it is now banned in several countries [10,11]. Glufosinate controls weeds by inhibiting glutamine synthetase (GS, EC 6.3.1.2) and accumulating ammonia, which is involved in the nitrogen metabolism pathway, ultimately causing plant death [12,13]. Additionally, an amino acid substitution of GS2 was found in a glufosinate-resistant annual grass (*Lolium perenne* L. spp. *Multiflorum*) [14]. Furthermore, the functions of GS in horseweed (*Conyza canadensis*), palmer amaranth (*Amaranthus palmeri*), johnsongrass (*Sorghum halepense*), kochia (*Kochia scoparia*), and ryegrass (*Lolium rigidum*) have been reported [15].

In the nitrogen metabolism pathway, the primary pathway of GS is an essential enzyme of nitrogen metabolism that assimilates ammonia in higher plants, requiring ATP [16]. The glutamine synthetase/[glutamate synthetase (glutamate 2-oxoglutarate aminotransferase), EC 1.4.1.13 and 1.4.1.14] (GS/GOGAT) cycle is the pathway responsible for the incorporation of inorganic nitrogen (N) into organic molecules. For example, GS2 is involved in the nitrogen deficiency of rice and ammonium tolerance in C_3_ plants [17,18,19]. Meanwhile, the minor pathway of glutamate dehydrogenase (GDH, EC 1.4.1.3) links the carbon and nitrogen metabolisms as another way to assimilate ammonia into glutamate or deaminate glutamate into 2-oxoglutarate and ammonia, requiring NADPH [17]. GDH acts as a shunt to the GS/GOCAT cycle, responding to the differing needs of cells for carbon and nitrogen compounds [20]. The gene *gdhA* of GDH has been reported to enhance glufosinate resistance in transgenic tobacco [21].

Despite recent efforts to understand the mechanism of glufosinate resistance, the patterns of the genes and proteins of the nitrogen mechanism in glufosinate-resistant goosegrass have not yet been reported. Goosegrass (*Eleusine indica* L. Gaertn.) is one of the worst weeds (a self-pollinating monoecious species) in the world, which can produce large quantities of seeds germinating at any time of the year and being carried in clusters [22]. The taller the goosegrass, the harder it is to be controlled by glufosinate [23]. And it was first found resistant to glufosinate in Malaysia in 2009 [24,25]. Subsequently, multi-resistance of goosegrass to both glufosinate and paraquat was found in Malaysia [26]. This study attempted to identify the field-evolved target site resistance genes and proteins in glufosinate-resistant goosegrass. It provides more information to enhance the understanding of nitrogen metabolism and its regulation in response to glufosinate stress, as well as the identification of putative genes for use as references in the marker-assisted breeding of transgenic crops.

## 2. Results

### 2.1. Resistant Phenotypes and Dose Responses to Glufosinate Herbicides

To calculate the resistance level of glufosinate in three biotypes of goosegrass, bioassay experiments were carried out by spraying gradient doses of glufosinate solution on goosegrass. The goosegrass phenotypes show visualized glufosinate resistance, despite some damage (Figure 1a). The goosegrasses NX and CS began to wither after spraying glufosinate up to 25 and 200 g a.i.ha^−1^, respectively. However, whole plants of the goosegrass AUS still grew well after spraying a high dosage of 6400 g a.i.ha^−1^ of glufosinate. Moreover, the dose-response curves of the goosegrass biotypes showed a significant difference in the resistance levels of glufosinate (Figure 1b). The GR_50_ (growth reduction to 50%) value of the NX biotype in response to glufosinate was 6.37 g a.i.ha^−1^. The higher resistance level of the CS biotype (GR_50_ at 41.08 g a.i.ha^−1^) and the AUS biotype (GR_50_ at 394.90 g a.i.ha^−1^) toward glufosinate was 6.45- and 61.99-fold compared to that of the NX goosegrass biotype.

### 2.2. Glufosinate-Resistant Effects on Ammonia Content

To evaluate the toxicity of glufosinate resistance in goosegrass, the standard curve of ammonium was established to determine the ammonium content in glufosinate-susceptible and -resistant goosegrass populations (Figure 2a). The ammonium content across the populations was similar before applying glufosinate (400 g a.i.ha^−1^), but the accumulation of ammonium significantly varied and increased with the treatment time (Figure 2b). At 0 h, the ammonium content of the NX, CS, and AUS goosegrass biotypes were 4.22, 2.62, and 3.13 mg/g, respectively. The ammonium content of the different goosegrass biotypes gradually increased and finally tended to become flat under glufosinate treatment. At 66 h, the accumulation of ammonium of the CS biotype (21.49 mg/g) and the AUS biotype (14.73 mg/g) in response to glufosinate was 0.53- and 0.77-fold compared to that of the NX biotype (27.91 mg/g). Thus, the higher the level of glufosinate resistance in goosegrass, the lower the ammonium accumulation in response to the stress of glufosinate.

### 2.3. Glufosinate-Resistant Effects on GS, GOGAT, and GDH Activities

To validate the pivotal routes for the mechanism of glufosinate resistance, the activity of the targeted enzymes GS, GOGAT, and GDH in glufosinate-susceptible and -resistant goosegrass populations was determined for 7 g a.i.ha^−1^ of glufosinate. The GS activity in the NX, CS, and AUS goosegrass biotypes was inhibited under the stress of glufosinate (Figure 3a). After the stress, the GS activity in the glufosinate-susceptible goosegrass (NX) remained at a low level, while the GS activity rebounded in the two glufosinate-resistant goosegrasses (CS and AUS). In particular, the GS activity in AUS goosegrass began at a high level of 96.01 U/g and could rebound to 80.18 U/g at 36 h, and the average value of GS activity in AUS goosegrass was 2.77-fold that of NX. Meanwhile, the GOGAT activity in the NX, CS, and AUS goosegrass biotypes also decreased under the stress of glufosinate (Figure 3b). GOGAT activity experienced the most significant decline in the glufosinate-susceptible goosegrass (NX) from 131.61 to 34.24 U/g by 60 h after stress, but the GOGAT activity in the glufosinate-resistant goosegrasses merely decreased from 110.21 to 48.15 U/g (CS) and from 89.88 to 22.47 U/g (AUS) by 48 h after stress. Moreover, the GDH activity in the NX, CS, and AUS goosegrass biotypes increased first and then decreased under stress (Figure 3c). Compared to the GDH activity at 0 h, the GDH activity in the glufosinate-susceptible goosegrass (NX) increased by 82.82% by 36 h with a more significant range (69.87–127.74 U/g). However, the GDH activity increased in AUS by 82.65% by 36 h (84.02–153.46 U/g) and in CS by 52.19% by 12 h (107.60–163.75 U/g).

Therefore, it is evident that there was a higher activity of GS but a lower activity of GOGAT based on the higher resistance of goosegrass in response to glufosinate stress. Their different changing trends indicate that a mechanism involving ammonia assimilation within the GS/GOGAT pathway might be affected by nitrogen metabolism. It was also observed that the activity of GDH increased with the increased resistance of goosegrass, though its changing trend did not exhibit an obvious increase and mainly increased first and then dropped. This separate GDH pathway may provide some alleviation against glufosinate stress.

### 2.4. Relative gdhA Gene Expression by RT-PCR

To verify the effect of the GDH-dependent route in glufosinate resistance, the relative expression of *gdhA* was determined in glufosinate-susceptible and -resistant goosegrasses (NX, CS, and AUS) with or without 400 g a.i.ha^−1^ of glufosinate stress (Figure 4). The relative expression of the *gdhA* gene demonstrated no significant differences in NX, CS, and AUS goosegrass without glufosinate stress. However, the relative expression in the three populations significantly decreased periodically with glufosinate stress. The *gdhA* gene expression in the three populations was significantly downregulated at 12 h and then upregulated at 24 h under glufosinate stress. Additionally, the significant low *gdhA* gene expression levels at 60 h were 16.24-, 3.52-, and 23.99-fold lower than that at 0 h. Generally, the relative expression of the *gdhA* gene in the three populations showed the same trend in the different treatment periods. With the increase in glufosinate resistance, the relative expression of the *gdhA* gene in the three populations from largest to smallest was as follows: NX, CS, and AUS.

### 2.5. Effects of Glufosinate on Transcriptome Characterizations

Based on the results of the targeted enzyme activity and *gdhA* expression mentioned above, the goosegrass transcriptome was analyzed at 0 and 36 h after spraying 7 g a.i.ha^−1^ of glufosinate to generally assess the expression levels of other genes and proteins responding to the stress of glufosinate. RNA libraries of six samples (CK-NX, CK-CS, CK-AUS, T-NX, T-CS, and T-AUS) were obtained by sequencing, with CK and T representing goosegrass (NX, CS, and AUS) at 0 or 36 h, respectively. Valid reads of all samples were obtained by de novo concatenation, and duplicated splicing sequences were removed to obtain 49,828 unigenes with 89,894 transcripts. Compared to NCBI-NR (NCBI non-redundant protein database), 27,326 unigenes were identified and annotated to six species with the highest number of the same unigenes (Figure 5a), which were *Setaria italica* (17.52%), *Panicum hallii* (15.72%), *Panicum miliaceum* (12.58%), *Sorghum bicolor* (12.58%), *Zea mays* (8.08%), and *Dichantheium oligosanthe* (7.33%). Meanwhile, there were differential genes in the comparisons involving glufosinate treatment and different resistant goosegrass populations. Under glufosinate treatment, the stronger the resistance, the fewer differential genes (Figure 5b). Specifically, the maximum number of differential genes occurred in the NX goosegrass (2452 genes), followed by the CS goosegrass (733 genes), while only the minimum was found in the glufosinate-resistant AUS goosegrass (515 genes), in response to the stress of glufosinate. Under H_2_O treatment, there were inherent differences in gene expression among the different resistant species of T-CS (3124 genes) and T-AUS (1206 genes) goosegrasses, compared to the glufosinate-susceptible goosegrass (NX). Moreover, only 34 genes showed distinct expression differences in all treatments and materials. Thus, we screened and found that 14 of these genes were annotated by NCBI-NR (Table 1). Furthermore, clustering analysis of the differential gene expression among the three goosegrass biotypes showed that the highly resistant goosegrass (AUS) exhibited significantly different expression patterns in its metabolism, regardless of whether it was subjected to glufosinate stress, in comparison to the other two goosegrass biotypes (Figure 6). The expression patterns between the glufosinate-susceptible goosegrass (NX) and the medium-resistant goosegrass (CS) were relatively similar. Among the cluster analysis, four goosegrass transcripts (TRINITY_DN22166_c1_g8, TRINITY_DN22456_c0_g8, TRINITY_DN22857_c1_g10, and TRINITY_DN8154_c0_g2) belonged to the range of 34 genes, and one gene (Os04g0617900, germin-like protein 4-1) fell into the category of 14 annotated genes. However, these genes identified through transcriptome screening were primarily essential genes rather than the specifically targeted genes related to glufosinate resistance. 

### 2.6. Effects of Glufosinate on Proteomic Characterizations

Based on the transcriptome data of goosegrass, six samples were further used for proteomic analysis, as per the transcriptome analysis. The proteomic sequencing identified 22,088 unique peptides with specific segments and 4783 proteins. The expression of the differential proteins in the different treatment groups was analyzed, and it was found that the expression of these proteins was more regular than that of the differential genes in goosegrass (Figure 7a). Additionally, the differential proteins of goosegrass under glufosinate treatment were mainly downregulated, which was confirmed from the number of up- or downregulated proteins in the different groups. Specifically, glufosinate-resistant goosegrass has the most downregulated proteins (T-CS vs. CK-CS and T-AUS vs. CK-AUS). Meanwhile, the smaller the difference in the level of resistance between two biotypes of goosegrass, the fewer differential genes between them under glufosinate treatment (T-CS vs. T-NX). This trend was consistent among goosegrass groups without glufosinate treatment (CK-AUS vs. CK-CS, CK-AUS vs. CK-NX, and CK-CS vs. CK-NX). Furthermore, the enrichment of KEGG pathways was analyzed in the nine comparative groups to obtain a more direct and systematic understanding of the mechanism of goosegrass under glufosinate stress (Appendix A) due to the need for differential proteins within organisms to coordinate the completion of a series of biochemical reactions in order to carry out their biological functions. Among the top 20 most significantly enriched KEGG pathways in all groups, the nitrogen metabolism was only enriched in the T-NX vs. CK-NX group (Figure 7b). This suggests that the nitrogen metabolism plays a critical role in the mechanism of glufosinate resistance between glufosinate-resistant and -susceptible goosegrass in response to glufosinate stress.

### 2.7. The Patterns of Expressed Genes and Proteins in the Nitrogen Metabolism

To further identify GS, GOGAT, and GDH in the nitrogen metabolism, we examined the relative expressions of the targeted proteins and genes in the nitrogen metabolism pathway detected by transcriptome and proteome analysis. The results showed that both transcription and protein expression levels of *GS*, *GOGAT*, *GDH*, *NirA*, *NR*, and *CA* in the nitrogen metabolism of the goosegrasses were enriched under glufosinate stress (Table 2). In addition, the possible transcript sequences of *GS*, *GOGAT*, *GDH*, *NirA*, *NR*, and *CA* in the nitrogen metabolism of goosegrass were identified according to the association analysis of transcriptome and proteome data (Table 3 and Table 4). After associating the majority of protein IDs with the corresponding transcript IDs of genes through unique peptides, consistent differences in fold changes were observed in the targeted proteins and genes, specifically *GS* (*fluG*, *GLN1-1*, *GLN1-2*, *GLN1-3*, and *GLN2*), *GDH* (*GDH* and *GDH2*), *GOGAT* (*Os01g0681900*, *Os05g0555600*, and *GLSF*), *NR* (*NIA1* and *CB5-A*), *NirA* (*Os01g0357100*), and *CA* (named *CA1*). Then, the putative resistance genes of goosegrass were screened out from a narrowed-down range responsible for glufosinate stress by combining the up- and downregulated changes in the enrichment of the KEGG pathway. The tables showed differences and similarities between two different biotypes of goosegrass (AUS and CS). On the one hand, the expression levels of genes and proteins in the GS family exhibit greater differences, such as 127.20-fold *GLN1-3* (1.75-fold GLN1-3) in CK-AUS/CK-CS and 1.97-fold *GLN1-1* (1.26-fold GLN1-1) in T-AUS/T-CS. On the other hand, it remained consistent that the expression levels of genes and proteins in the GS family had the highest expression level in the five families, especially compared to the NirA family. The highest expression levels were found in the five genes (Bold mark), with higher expression levels than other genes of each family in both AUS and CS goosegrasses. They were *GLN1-1*, *GDH2*, *GLSF*, *NIA1*, *Os01g0357100*, and *CA1*, with their CDS sequences shown in the Appendix A (Appendix A). Therefore, these genes may provide more specific information about glufosinate resistance for future functional validation of resistant genes and the construction of mutants.

## 3. Discussion

A graphic model of the differential genes and proteins in the nitrogen metabolism pathways under glufosinate stress in susceptible and resistant goosegrasses (NX, CS, and AUS) was analyzed by KEGG enrichment (Figure 8). The model visualizes the changes in differential expression in the nitrogen metabolism pathways in the same way as their original pathways (Appendix A). Generally, the transcription levels of all six genes (*GS*, *GOGAT*, *GDH*, *NirA*, *NR*, and *CA*) were affected in susceptible goosegrasses (T-NX vs. CK-NX) with *GDH* upregulated, including their protein levels of GDH, NR, and CA. However, glufosinate stress played a minor role in the nitrogen metabolism of resistant goosegrasses (T-AUS vs. CK-AUS), with only the transcription levels of *GS* being upregulated. This suggests that the glufosinate-resistant mechanism patterns in goosegrass still ultimately depend on changes in *GS*, which implies the importance of conducting further research on GS in glufosinate-resistant goosegrass (AUS) compared to susceptible goosegrass (NX). 

AUS and CS are two biotypes of *Eleusine indica* L. resistant to glufosinate. During goosegrass tolerance to glufosinate stress, AUS and CS employ distinct mechanisms of molecular regulation in the nitrogen metabolism. Specifically, different mechanisms for glufosinate resistance may exist. First, in the AUS goosegrass biotype, glufosinate is reduced by releasing ammonium accumulation through the upregulation of the transcription targeted enzyme GS. Second, the nitrogen source is cut off by adjusting the transportation of nitric acid and nitrite reductase to nitrogen to prevent the synthesis of ammonium, thereby reducing the toxicity of ammonium and generating resistance in the CS goosegrass biotype. Third, through the double action of GDH, the toxic effect of ammonium is alleviated to some extent. Fourth, carbonic anhydrase reduces ammonium accumulation in goosegrass by decreasing the respiratory ammonium produced during the photosynthetic process. Meanwhile, bioinformatics analysis of the transcriptome and proteomics data for the targeted genes indicates that the nitrogen metabolic pathway plays a critical role in the glufosinate resistance of goosegrass. The identified targeted genes are likely involved in the high resistance of goosegrass to glufosinate. However, this study only examined the expression level of candidate genes and did not address the resistance caused by targeted mutations, although their influence cannot be ruled out. Thus, further research into gene mutations in glufosinate-resistant goosegrass is still needed.

Different mechanisms for the resistance to glufosinate in other glufosinate-resistant species showed that the function of the bar gene was involved with penetration on resistance of a *Triticum aestivum* line [27]. And the Italian ryegrass (*Lolium perenne* ssp. *multiflorum*) was resistant to glufosinate with one amino acid substitution in GS2 but still poorly understood physiological and genetic mechanisms [8,28,29,30]. It was followed by a report of glufosinate-resistant rigid ryegrass (*Lolium rigidum* Gaud.) in Greece [31]. Meanwhile, RNA-Seq transcriptome analysis of Palmer amaranth (*Amaranthus palmeri*) was carried out to screen candidate genes related to glufosinate tolerance, such as P450 genes [13]. The chloroplastic glutamine synthetase enzyme (GS2) of resistant Palmer amaranth responded to glufosinate resistance by enhancing amplification and expressions without mutations [32]. The activity of protoporphyrinogen oxidase (PPO) inhibitors and the generation of reactive oxygen species (ROS) were related to the action mechanism of glufosinate, including in the species of Palmer amaranth [10,11,15,33]. Long-term research in the control of goosegrass in glufosinate-resistant cotton found that goosegrass tolerance to glufosinate may be caused by translocation limitation for most of the radioactivity retained in the leaves at about 90% [34,35,36,37]. Recently, a novel point mutation was found in the goosegrass mutant EiGS1-1 with a Ser59Gly substitution [9]. Taken overall, the effects of biological nitrogen metabolism on glufosinate-susceptible and -resistant goosegrass would play a role in future research for mechanisms of glufosinate-resistance in terms of the metabolic aspects.

In this study, the targeted genes *GLN1-1*, *GDH2*, *GLSF*, *NIA1*, *Os01g0357100*, and *CA1* were selected from goosegrass. These genes have recently been reported to have various functions in other plants. For example, in rice germination, inorganic N (free N ions, nitrate, and ammonium) is absorbed, and ammonium is combined with glutamate to form glutamine through the catalysis of GS (*GS* or *GLN*) [38]. Nitrate reductase 1 (NIA1) and NIA2 in *Arabidopsis thaliana* plants play a role in assimilating nitrate into ammonia [39]. The enzyme carbonic anhydrase (CA, EC 4.2.1.1) is considered to promote the interconversion of bicarbonate with carbon dioxide (CO_2_) and water in C_3_ plants [40]. Therefore, the resistance genes identified in this study are interconnected with genes responsible for high nitrogen utilization. Plants possess genes that fulfill multiple functions, including plant growth and stress resistance, simultaneously. Meanwhile, adopting diverse weed management methods through crop rotation would be the most suitable approach for controlling herbicide-resistant weeds [7].

In conclusion, this study clarified the nitrogen metabolism pathway that influences the glufosinate-resistant goosegrass in response to stress. The putative target genes *GLN1-1*, *GDH2*, *GLSF*, *NIA1*, *Os01g0357100*, and *CA1* were screened out. The expression patterns of the putative genes involved in glufosinate-resistant goosegrass could serve as a basis for future functional verification in transgenic plants.

## 4. Materials and Methods

### 4.1. Plant Materials

Natural goosegrass seed samples were collected from the following areas: Ningxi Field (in the town of Ningxi in the district of Zengcheng, Guangzhou, Guangdong, China) (NX, E113°49′N23°08′, originally collected in 2015; purified in South China Agricultural University, E113°36′N23°16′), the town of Shaxi (Chaozhou, Chaoshan Area, Guangdong, China) (CS02, E166°38′N23°41′, originally collected in 2015; purified in South China Agricultural University, E113°36′N23°16′), and the University of Western Australia (Perth, Australia) (AUS, purified in Perth, E115°52′S31°52′; originally collected in the Jerantut farm of Malaysia in 2013, E102°22′N3°56′). The trials were carried out in the greenhouses and laboratories of South China Agricultural University, Guangzhou, Guangdong Province, China. Seeds of the three goosegrass biotypes (NX, CS, and AUS) stored in the refrigerator (4 °C) were manually removed from the shells, soaked in a solution of gibberellin (5 mg/L) for 24 h to break dormancy, and then sowed in seedling pots (15 × 25 cm) with nutrient soil and a small amount of sand (2:1, *v*/*v*). Goosegrass plants of the three biotypes were placed in a growth chamber (Guangzhou Shenhua Biotechnology Co., Ltd., Guangzhou, China) maintained at 30/27 °C day/night temperature (80 µmol/m^2^·s, 80% relative humidity) with a 12 h photoperiod. The plants were transplanted into pots (with a diameter of 9 cm) at the 2–3 leaf stage, with each pot containing five seedlings.

### 4.2. GR_50_ of the Glufosinate Treatments

More than 15 plants (NX, CS, and AUS) at the 5–6 leaf stage per accession were separately sprayed by a 3WP-2000 spray tower (Nanjing Research Institute for Agricultural Mechanization, Ministry of Agriculture, Nanjing, China) with an 18% glufosinate solution (Basta 200SL, Bayer, Germany) in gradients of 12.5, 25, 50, 100, 200, 400, 800, 1600, 3200, and 6400 g of active ingredient (a.i.) ha^−1^. The weight of the aboveground fresh biomass was measured and photographs taken 14 days after spraying [41,42,43]. Regression analysis was conducted on the glufosinate dosage *x* and inhibition rate of the fresh-weight *y* using SPSS 17.0 and Originpro 8.5.0 software (*y* = a ln[−b ln(*x*)]). The dose-inhibition rate fitting curve, value of GR_50_ (50% inhibition of weed growth), confidence intervals, and correlation coefficients and relative folds of glufosinate resistance in the goosegrass (NX, CS, and AUS) were also calculated.

### 4.3. Ammonium Content

The goosegrass biotypes (AUS, CS, and NX) at the six-leaf stage were sprayed with 400 g a.i.ha^−1^ of glufosinate (the highest lethal concentration of the NX goosegrass biotype mentioned above). Leaves of the goosegrasses were collected 0, 12, 24, 36, 48, and 60 h after glufosinate spraying, with ddH_2_O as a control and repeating each treatment four times, before storage at −80 °C for testing.

The leaves (0.25 g) were ground with liquid nitrogen, extracted with a polyethylene solution (50 mg of polyethylene in 1 mL of ddH_2_O), and then centrifuged at 12,000× *g* at 4 °C for 5 min. The supernatants (200 μL) were diluted with ddH_2_O (800 μL), and 20 μL of reagent A was added (5 g of C_6_H_5_OH, 25 mg of Na_2_[Fe(CN)_5_NO]·2H_2_O, and 500 mL of ddH_2_O), followed by 1.5 mL of reagent B (2.5 g of NaOH, 1.6 mL of NaOCl, and 500 mL of ddH_2_O) after shaking evenly. The mixtures were incubated at a constant temperature of 37 °C for 15 min and the absorbance was measured at a wavelength of 625 nm using a UV-2550 ultraviolet spectrophotometer.

### 4.4. qRT-PCR

The relative expression levels of the *gdhA* gene in the goosegrass leaves were analyzed in the same materials used for measuring the ammonium content, which were taken at 0, 12, 24, 36, 48, and 60 h after glufosinate spraying. The 18 goosegrass leaf samples (50–100 mg) were ground in liquid nitrogen. RNA extraction from these samples was carried out using the RNA extraction kit EasyPure^@^ Plant RNA Kit (Code No. ER301-01, TransGen Biotech Co., Ltd., Beijing, China). Reverse transcription and synthesis of cDNA were carried out by referring to the instructions of the TransScript^®^ II One-Step gDNA Removal and cDNA Synthesis SuperMix reagent kit (Code No. AH311-02, TransGen Biotech Co., Ltd., Beijing, China). The cDNA of the samples was stored at 20 °C for RT-PCR use. The transcript sequence of the *gdhA* gene was obtained from the transcriptome results of our previous research by BLAST similarity analysis of the nucleotide sequence on the NCBI (https://www.ncbi.nlm.nih.gov/ (accessed on 28 February 2023) website. The RT-PCR primer for the *gdhA* gene designed by Primer 5.0 was gdhA-F/R (5′-CAGTTCAGTCAGGCATTA-3′/5′-AACGCATTATCTCATTATCAC-3′), with Actin-F/R (5′-AACATCGTTCTCAGTGGTGG-3′/5′-CCAGACACTGTACTTCCTTTCA-3′) as the primer of an inner reference gene from the goosegrass [44,45,46,47]. The cDNA template (2 µL) was used with the RealMaster Mix SYBR Green (Code No. FP202, TIANGEN Biotech (Beijing) Co., Ltd., Beijing, China). qRT-PCR reactions were performed under the following conditions: 50 °C for 2 min, 95 °C for 2 min, 40 cycles at 95 °C for 15 s, and 60 °C for 1 min, followed by 95 °C for 15 s, 60 °C for 1 min, and 95 °C for 30 s for dissociation curve analysis (three replicates).

### 4.5. Assay Enzyme Activity of GS, GOGAT, and GDH

The goosegrass biotypes (AUS, CS, and NX) at the six-leaf stages were sprayed with 7 g a.i.ha^−1^ of glufosinate (GR_50_ of NX goosegrass mentioned above). Leaves of the goosegrasses (0.1 g) were also collected 0, 12, 24, 36, 48, and 60 h after glufosinate spraying, with ddH_2_O as the control and repeating each treatment four times, before storage at −80 °C for testing.

Extraction of the GS, GOGAT, and GDH enzymes was carried out using Solarbiogo’s reagent kit (Beijing Solarbio Science & Technology Co., Ltd., Beijing, China) following the manufacturer’s instructions. The activity of the GS and GOGAT enzymes was measured at absorption values of 540 nm (BC0915-100T/48S) and 340 nm (BC0075-100T/96S), respectively, using a microplate reader. The activity of GDH was measured at the absorption value of 340 nm (BC1460-50T/48S) using a UV-2550 ultraviolet spectrophotometer.

### 4.6. Illumina Sequencing and iTRAQ-TMT Proteome Analysis

The transcriptome and proteome analysis materials were selected from the same materials used for the enzyme activity assay, which were collected 36 h after glufosinate spraying, with 0 h as the control. All samples were mixed into the six groups of CK-NX, CK-CS, CK-AUS, T-NX, T-CS, and T-AUS for transcriptome sequencing (Illumina Novaseq™ 6000) and proteome sequencing (iTRAQ, isobaric Tags for Relative and Absolute Quantitation), which was performed by LC-Bio Technologies (Hangzhou) Co., Ltd. (Hangzhou, China) following the vendor’s recommended protocol (LC Sciences, Houston, TX, USA).

In brief, the quantity and purity of the total RNA in the six goosegrass samples were analyzed using Bioanalyzer 2100 and an RNA 1000 Nano LabChip Kit (Agilent, CA, USA) with RIN number > 7.0, after extraction by Trizol reagent (Invitrogen, CA, USA). Two rounds of purification of poly(A) RNA from the total RNA (5 μg) were conducted with poly-T oligo-attached magnetic beads. The cleaved fragments of mRNA, broken down into small pieces by divalent cations, were reverse-transcribed to form the final cDNA library using an mRNASeq sample preparation kit (Illumina, San Diego, CA, USA). An average insert size of 300 bp (±50 bp) was used for the paired-end libraries, which were subjected to paired-end sequencing on an Illumina Novaseq™ 6000. Moreover, the quantities of total protein in the six goosegrass samples were measured using a BioDrop μLite micro-detector (BioDrop, Biochrom Ltd., Cambridge, UK) and sodium dodecyl sulfate-polyacrylamide gel electrophoresis (SDS-PAGE). After the total protein (100 μg) had been digested by Sequencing Grade Modified Trypsin, the peptides were desalted and reconstituted in 40 μL of TEAB (100 mM) using an 8-plex iTRAQ kit (AB sciex Inc., Framingham, MA, USA). Then, the labeled peptides were analyzed with the Nano-LC-ESI-MS/MS system, fractionated by a BEH chromatographic column (C18, 5 μm, 4.6 × 250 mm, Waters Inc, Milford, MA, USA) and passed through EASY-nLC 1000 coupled to a Q Exactive mass spectrometer (C18, 3 μm, 75 × 250 mm, Thermo Scientific, Austin, TX, USA).

### 4.7. Bioinformatics Analyses

The following bioinformatics analyses were carried out:

De novo assembly, unigene annotation, and functional classification. The read sequences of high quality were verified by FastQC (http://www.bioinformatics.babraham.ac.uk/projects/fastqc/ (accessed on 28 February 2023) to gain the Q20, Q30, and GC content of the clean data for downstream analyses. Then, de novo assembly of this transcriptome was carried out using Trinity 2.4.0. Trinity group transcripts into clusters based on shared sequence content, with the per transcript cluster loosely referred to as a “gene”. Among the clusters, the longest transcript was chosen as the “gene” sequence, which was defined as the unigene. All unigenes were assembled using DIAMOND with an E-value threshold of <0.00001 to align against databases, including the non-redundant (Nr) protein database (http://www.ncbi.nlm.nih.gov/ (accessed on 28 February 2023), Gene Ontology (GO) (http://www.geneontology.org (accessed on 28 February 2023), SwissProt (http://www.expasy.ch/sprot/ (accessed on 28 February 2023), Kyoto Encyclopedia of Genes and Genomes (KEGG) (http://www.genome.jp/kegg/ (accessed on 28 February 2023), and eggNOG (http://eggnogdb.embl.de/ (accessed on 28 February 2023).

Differential unigene analysis. Salmon was used to determine the expression level of the unigenes by calculating TPM. The differential unigenes were selected with a log2 (fold change) >1 or <−1 and with statistical significance (*p* < 0.05) using the R package edgeR. GO and KEGG enrichment analysis was performed on the differential unigenes using perl scripts in house.

Protein identification and quantification. MaxQuant (version 1.5.5.1) software was used to search the putative protein database of UniProt based on the transcriptome of the goosegrasses. The intensity value was obtained from protein databases as quantitative data, and a comparative analysis of the differences between the two groups was conducted after normalizing the intensity data.

Differential proteins analysis. The biological and functional properties of the proteins were analyzed using the GO and KEGG databases. A hypergeometric test was conducted to find significantly enriched GO terms and KEGG pathways of the proteins to define the significant enrichment of differential proteins (*p* < 0.05). The Clusters of Orthologous Groups of Proteins System was used for functional classification of the differential proteins.

### 4.8. Statistical Analysis

The data of the physiological parameters and qPCR analysis showed significant differences, which were analyzed by analysis of variance (ANOVA) and Duncan’s test with DPS 7.05 (Hangzhou, China), with different letters indicating significant differences (*p* < 0.05). Figures were prepared using Excel 2016, OriginPro 8.5.0 (OriginLab Corporation, Northampton, MA, USA), GraphPad Prism 8.01 (GraphPad Software, LLC, San Diego, CA, USA), and Adobe Illustrator CS4 (Adobe Systems Incorporated, San Jose, CA, USA).

## Figures and Tables

**Figure 1 ijms-24-13791-f001:**
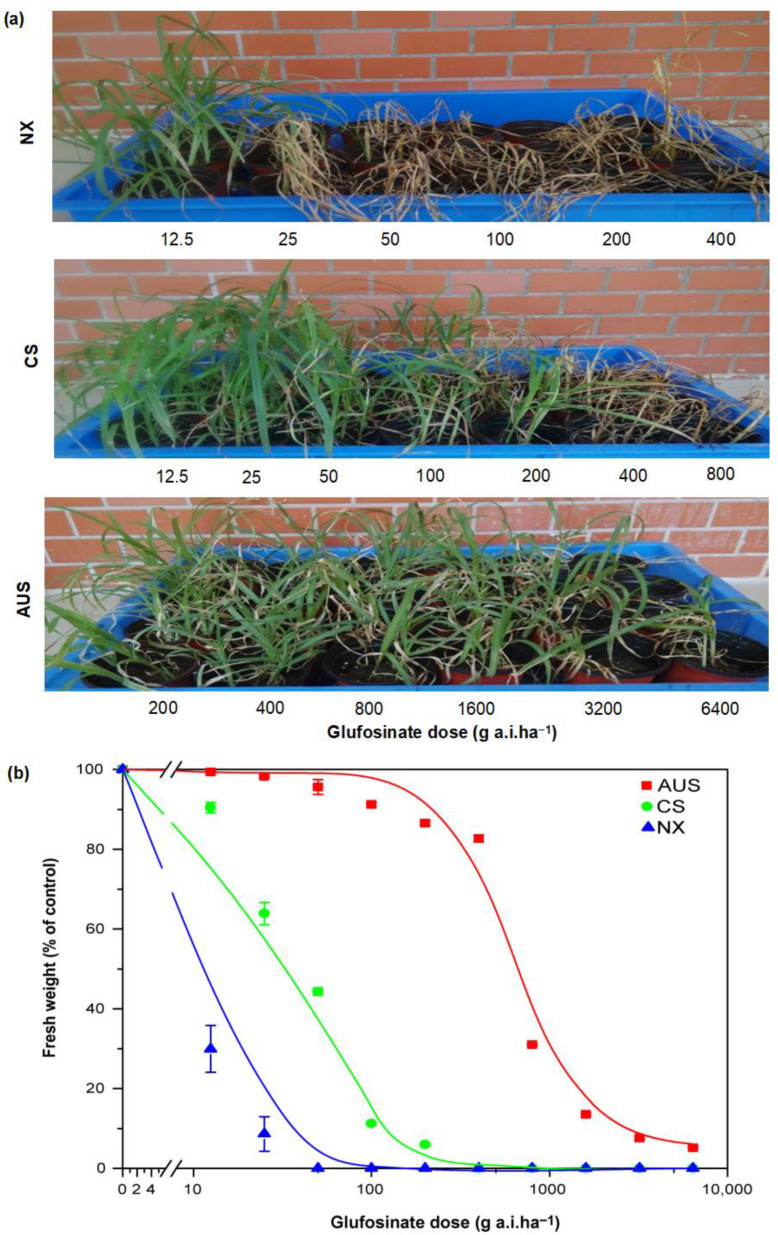
The resistance level of glufosinate in goosegrass populations. (**a**) Glufosinate-resistant phenotypes of goosegrass at 14 days after spraying gradient doses of glufosinate. NX, CS, and AUS refer to the goosegrass biotypes. (**b**) Effect of glufosinate doses on the fresh weight of goosegrass at 14 days. Three replicates.

**Figure 2 ijms-24-13791-f002:**
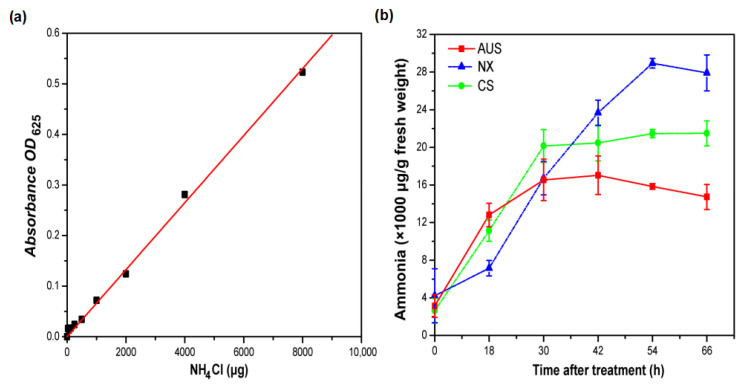
Ammonia content. (**a**) The standard curve of ammonia. (**b**) Effects of glufosinate (400 g a.i.ha^−1^) on the content of ammonia in glufosinate-susceptible and -resistant populations. NX represent glufosinate-susceptible goosegrass biotype; CS and AUS represent glufosinate-resistant goosegrass biotypes. Three replicates.

**Figure 3 ijms-24-13791-f003:**
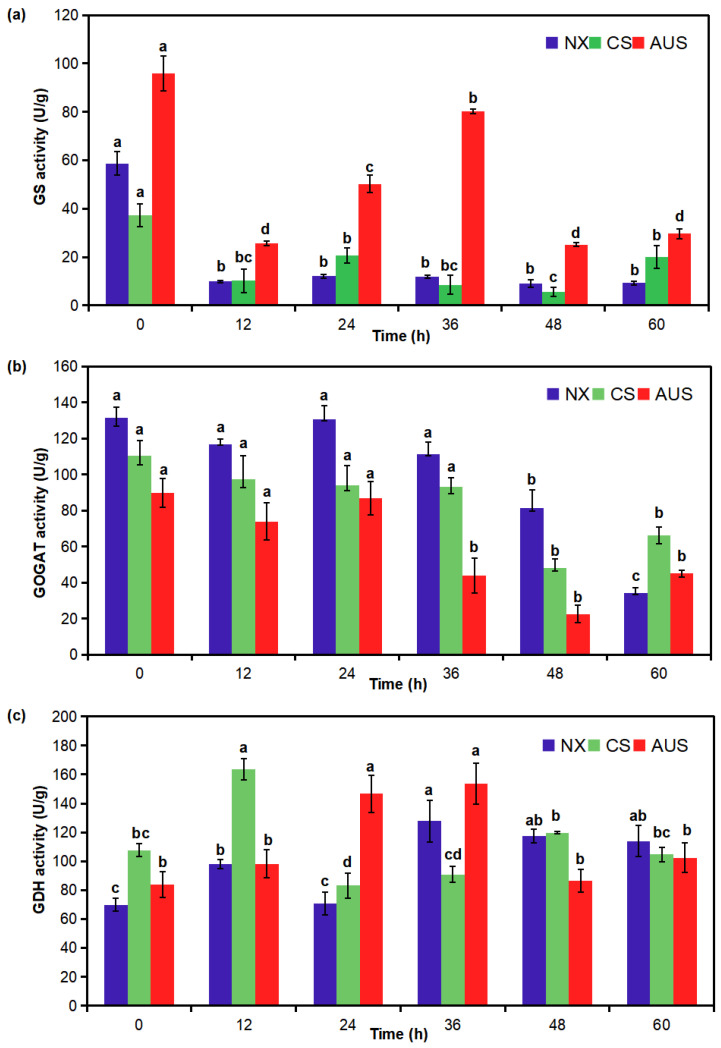
Effect of glufosinate resistance on GS, glutamine synthetase (**a**); GOGAT, glutamate 2-oxoglutarate aminotransferase (**b**); and GDH, glutamate dehydrogenase (**c**) activity in glufosinate-susceptible and -resistant goosegrasses with or without glufosinate stress (7 g a.i.ha^−1^). NX represent glufosinate-susceptible goosegrass biotype; CS and AUS represent glufosinate-resistant goosegrass biotypes. Different letters indicate significant differences *(p <* 0.05). Three replicates.

**Figure 4 ijms-24-13791-f004:**
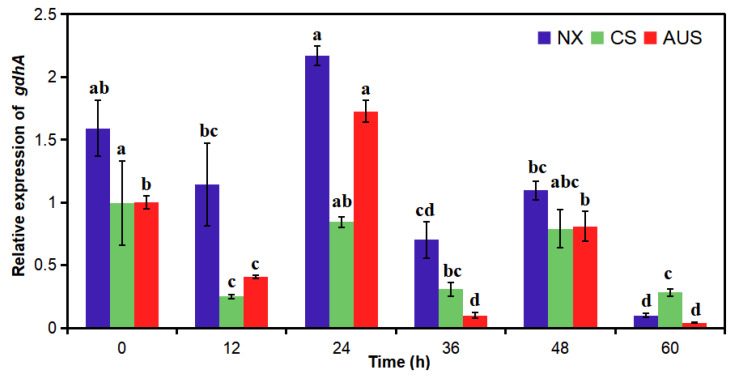
Relative expression of the *gdhA* gene in glufosinate-susceptible and -resistant goosegrasses with or without glufosinate stress (400 g a.i.ha^−1^). NX represent glufosinate-susceptible goosegrass biotype; CS and AUS represent glufosinate-resistant goosegrass biotypes. Different letters indicate significant differences *(p* < 0.05). Three replicates.

**Figure 5 ijms-24-13791-f005:**
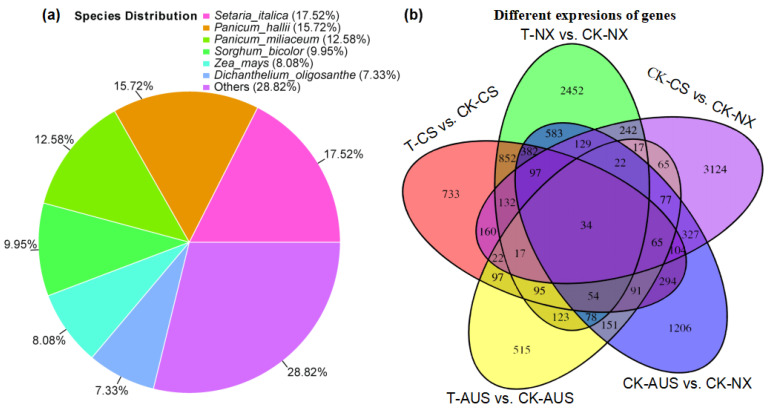
General analysis of the characterizations of glufosinate-resistant goosegrass by transcriptome sequencing. (**a**) Species distribution of goosegrass. (**b**) Differential genes in goosegrass under glufosinate treatment. T-NX, T-CS, and T-AUS represent the different goosegrass biotypes (NX, CS, and AUS) under glufosinate treatment, respectively. CK-NX, CK-CS, and CK-AUS represent the different goosegrass biotypes (NX, CS, and AUS) under H_2_O as the controls.

**Figure 6 ijms-24-13791-f006:**
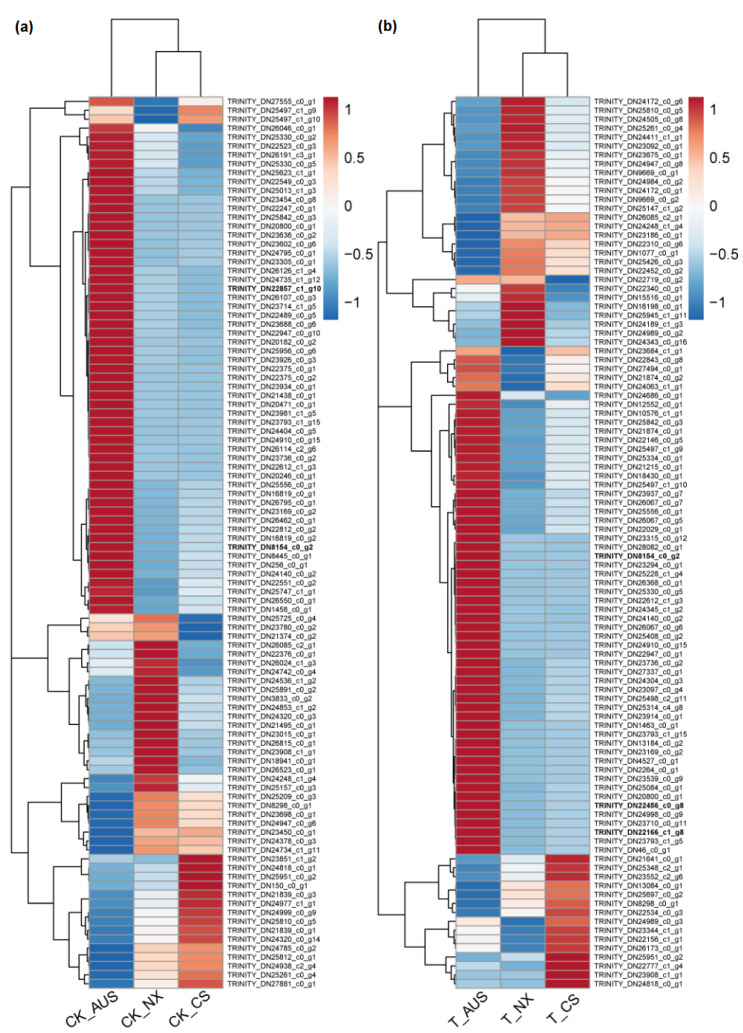
Heat map based on cluster analysis of significantly differential genes in goosegrass. (**a**) CK-NX, CK-CS, and CK-AUS represent the different goosegrass biotypes (NX, CS, and AUS) under H_2_O as the controls. (**b**) T-NX, T-CS, and T-AUS represent the different goosegrass biotypes (NX, CS, and AUS) under glufosinate treatment, respectively. Different colors represent different levels of gene expression.

**Figure 7 ijms-24-13791-f007:**
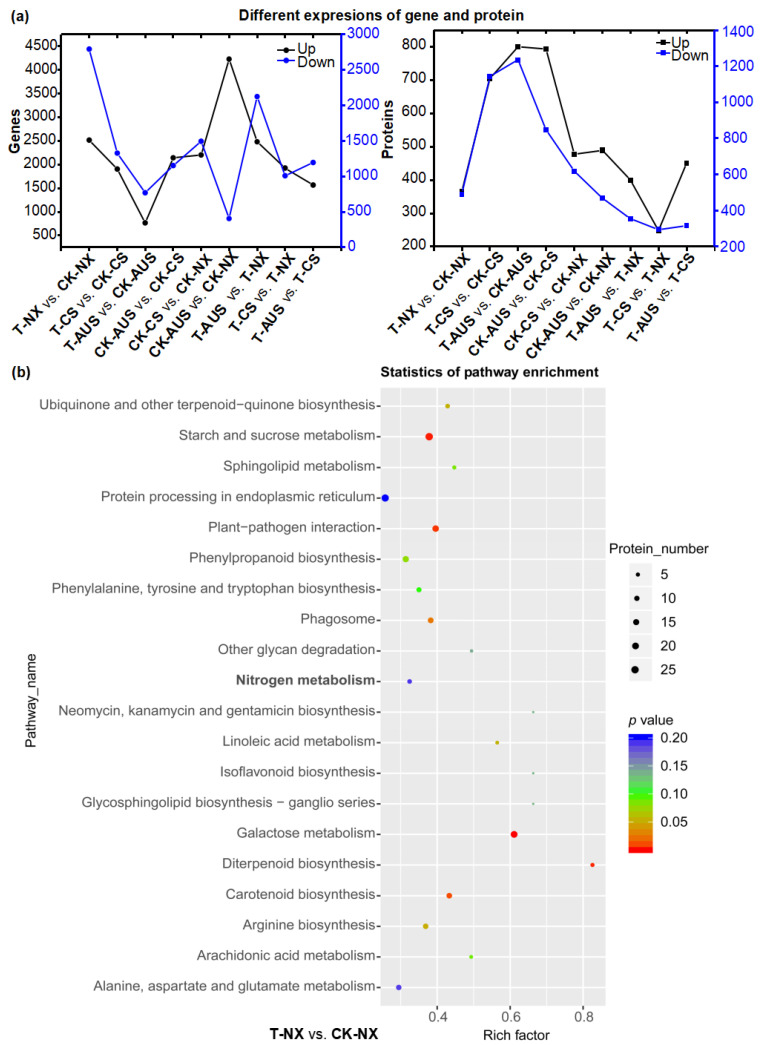
General analysis of the characterizations of glufosinate-resistant goosegrass by proteomic sequencing. (**a**) Correspondence analysis of differential genes and proteins under the comparative groups of glufosinate-resistant goosegrass and glufosinate treatment. (**b**) Enrichment scatter plot of KEGG pathways in the T-NX vs. CK-NX group.

**Figure 8 ijms-24-13791-f008:**
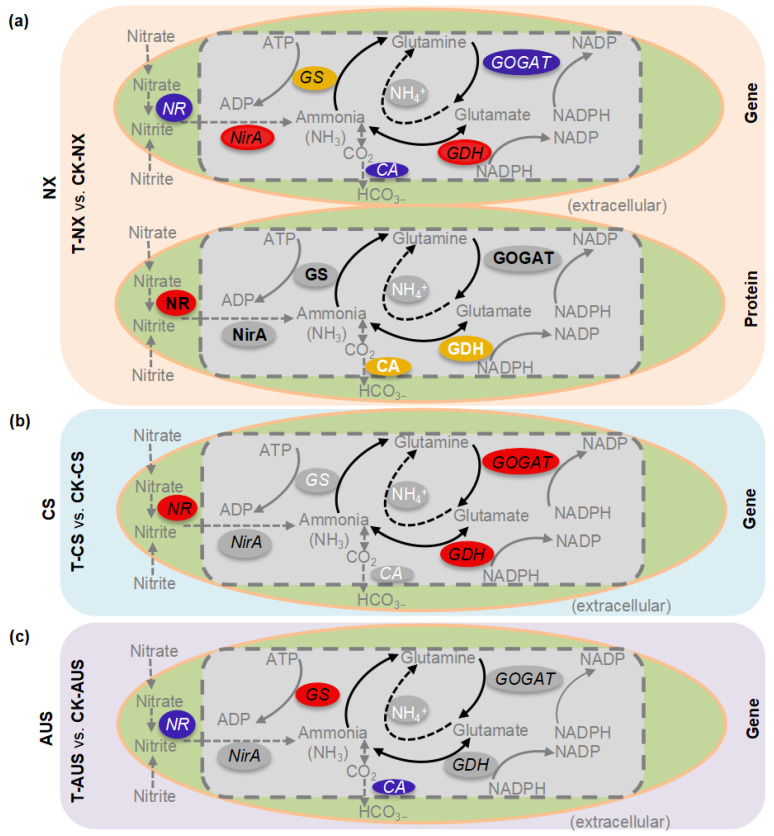
Graphic models of nitrogen metabolism pathways in response to glufosinate stress in susceptible and resistant goosegrass (NX, CS, and AUS). (**a**) Transcriptome and proteome analyses in the comparative group T-NX vs. CK-NX. (**b**,**c**) Transcriptome analyses in the comparative group T-CS vs.CK-CS and T-AUS vs. CK-AUS, respectively. Different colors mean differential genes and proteins annotated by the transcriptome or proteome, such as red (upregulated), blue (downregulated), yellow (both up- and downregulated), and grey (non-significantly regulated).

**Table 1 ijms-24-13791-t001:** Annotation information of the 14 genes screened from the 34 genes expressed in all materials and treatments of glufosinate resistance.

Gene	Annotation (Species)	EC	KEGG_Pathway
*CYP85A1*	Hypothetical protein GQ55_9G165800, *Panicum hallii*	EC:1.14.-.-	sita00906 (Brassinosteroid biosynthesis)
*MLO1*	MLO-like protein 1, *Dichanthelium oligosanthes*	NA	NA
*CER3*	Protein ECERIFERUM 3-like, *Panicum hallii*	EC:4.1.99.5	smo00074 (Cutin, suberine, and wax biosynthesis)
*GA20ox1B*	Gibberellin 20 oxidase 1-D-like, *Panicum miliaceum*	EC:1.14.11.12	sbi00905 (Diterpenoid biosynthesis)
*Os04g0617900*	Germin-like protein 4-1, *Setaria italica*	NA	NA
*Cht8*	Chitinase 8, *Setaria italica*	EC:3.2.1.14	sita00520 (Amino sugar and nucleotide sugar metabolism)
*Os04g0339400*	Probable aldo-keto reductase 3, *Setaria italica*	EC:1.1.1.65	cre00750 (Vitamin B7 metabolism)
*RPM1*	Hypothetical protein GQ55_2G362800, *Panicum hallii*	NA	bdi04627 (Plant–pathogen interaction)
*PEP1*	Phosphoenolpyruvate carboxylase, *Chloris gayana*	EC:4.1.1.31	sita00620 (Pyruvate metabolism)
*At3g16150*	Probable isoaspartyl peptidase/L-asparaginase 2, *Panicum hallii*	NA	NA
*LKR/SDH*	Alpha-aminoadipic semialdehyde synthase isoform X1, *Setaria italica*	EC:1.5.1.8 1.5.1.9	sita00311 (Lysine degradation)
*Os03g0733400*	Zinc finger BED domain-containing protein ricesleeper 2-like, *Setaria italica*	EC:3.4.19.12	bna04145 (Endocytosis)
*At5g08350*	GEM-like protein 4, *Panicum miliaceum*	EC:2.3.2.32	gmx04141 (Protein processing in endoplasmic reticulum)
*LHCA4*	Chlorophyll a-b binding protein 4, chloroplastic, *Sorghum bicolor*	NA	sbi00197 (Photosynthesis-antenna proteins)

**Table 2 ijms-24-13791-t002:** Annotation information of the targeted proteins and genes in the nitrogen metabolism pathway.

Protein	Majority Protein ID	Unique Peptides	Gene	Length	EC
GS	Gene.4172::TRINITY_DN15902_c0_g2::g.4172::m.4172	7	*fluG*	2716	EC:6.3.1.2
	Gene.49488::TRINITY_DN25503_c1_g2::g.49488::m.49488	13	*GLN1-1*	1414	EC:6.3.1.2
	Gene.49499::TRINITY_DN25503_c1_g6::g.49499::m.49499	13	*GLN1-2*	1371	EC:6.3.1.2
	Gene.49492::TRINITY_DN25503_c1_g4::g.49492::m.49492	2	*GLN1-3*	1487	EC:6.3.1.2
	Gene.51130::TRINITY_DN25703_c0_g2::g.51130::m.51130	10	*GLN2*	2201	EC:6.3.1.2
GDH	Gene.12781::TRINITY_DN20596_c0_g4::g.12781::m.12781	10	*GDH1*	1508	EC:1.4.1.3
	Gene.24624::TRINITY_DN23079_c2_g5::g.24624::m.24624	8	*GDH2*	2083	EC:1.4.1.3
GOGAT	Gene.45299::TRINITY_DN25064_c1_g1::g.45299::m.45299	25	*Os01g0681900*	7729	EC:1.4.1.13 1.4.1.14
	Gene.45304::TRINITY_DN25064_c1_g2::g.45304::m.45304	11	*Os05g0555600*	6655	EC:1.4.1.13 1.4.1.14
	Gene.32477::TRINITY_DN23865_c0_g14::g.32477::m.32477	49	*GLSF*	5055	EC:1.4.7.1
NR	Gene.41212::TRINITY_DN24663_c0_g1::g.41212::m.41212	5	*NIA1*	3174	EC:1.7.1.1 1.7.1.2 1.7.1.3
	Gene.51843::TRINITY_DN25791_c1_g6::g.51843::m.51843	1	*CB5-A*	879	EC:1.7.1.1 1.7.1.2 1.7.1.3
NirA	Gene.11975::TRINITY_DN20338_c0_g2::g.11975::m.11975	19	*Os01g0357100*	2060	EC:1.7.7.1
CA	Gene.23527::TRINITY_DN22941_c0_g1::g.23527::m.23527	12	*CA1*	932	EC:4.2.1.1
	Gene.23538::TRINITY_DN22941_c0_g5::g.23538::m.23538	4	*-*	534	EC:4.2.1.1
	Gene.42475::TRINITY_DN24781_c0_g1::g.42475::m.42475	1	*-*	860	EC:4.2.1.1

**Table 3 ijms-24-13791-t003:** Relative expression of the targeted proteins and genes in the nitrogen metabolism pathway detected by proteome analysis.

Proteome		Norm						Fold					
		CK_NX	CK_CS	CK_AUS	T_NX	T_CS	T_AUS	CK_CS/ CK_NX	CK_AUS/CK_NX	CK_AUS/CK_CS	T_CS/ T_NX	T_AUS/ T_NX	T_AUS/ T_CS
GS	fluG	0.51	0.60	0.57	0.52	0.48	0.54	1.17	1.11	0.94	0.93	1.04	1.11
	**GLN1-1**	**5.81**	**5.98**	**6.28**	**6.99**	**5.31**	**6.69**	**1.03**	**1.08**	**1.05**	**0.76**	**0.96**	**1.26**
	GLN1-2	2.46	2.78	3.63	3.05	2.99	3.43	1.13	1.47	1.31	0.98	1.12	1.15
	GLN1-3	0.08	0.10	0.17	0.12	0.12	0.12	1.23	2.15	**1.75**	1.00	0.97	0.97
	GLN2	1.99	2.30	2.57	2.26	2.70	2.86	1.16	1.29	1.12	1.20	1.27	1.06
GDH	GDH1	2.77	3.55	2.25	2.30	2.26	1.87	1.28	0.81	0.63	0.98	0.81	0.82
	**GDH2**	**2.04**	**1.85**	**2.09**	**1.91**	**1.49**	**1.58**	**0.91**	**1.03**	**1.13**	**0.78**	**0.83**	**1.06**
GOGAT	Os01g0681900	4.72	3.50	3.54	4.72	4.31	5.27	0.74	0.75	1.01	0.91	1.12	1.22
	Os05g0555600	0.48	0.37	0.50	0.51	0.46	0.57	0.78	1.03	1.32	0.91	1.12	1.23
	**GLSF**	**9.99**	**8.89**	**7.99**	**9.39**	**9.79**	**10.61**	**0.89**	**0.80**	**0.90**	**1.04**	**1.13**	**1.08**
NR	**NIA1**	**0.23**	**0.20**	**0.20**	**0.29**	**0.33**	**0.30**	**0.87**	**0.87**	**1.00**	**1.13**	**1.01**	**0.90**
	CB5-A	0.36	0.37	0.41	0.35	0.34	0.38	1.04	1.16	1.11	0.97	1.09	1.13
NirA	**Os01g0357100**	**2.56**	**1.44**	**1.63**	**2.70**	**3.74**	**3.16**	**0.56**	**0.63**	**1.13**	**1.38**	**1.17**	**0.84**
CA	**CA1**	**10.38**	**10.78**	**10.25**	**12.77**	**14.23**	**11.23**	**1.04**	**0.99**	**0.95**	**1.11**	**0.88**	**0.79**
	-	1.52	1.90	1.69	1.41	1.28	1.18	1.25	1.11	0.89	0.91	0.83	0.92
	-	0.66	0.60	0.53	0.83	1.15	0.84	0.90	0.80	0.90	1.38	1.01	0.73

**Table 4 ijms-24-13791-t004:** Relative expression of the targeted proteins and genes in the nitrogen metabolism pathway detected by transcriptome analysis.

Transcriptome		TPM						Fold					
		CK_NX	CK_CS	CK_AUS	T_NX	T_CS	T_AUS	CK_CS/ CK_NX	CK_AUS/ CK_NX	CK_AUS/ CK_CS	T_CS/ T_NXz	T_AUS/ T_NX	T_AUS/ T_CS
GS	*fluG*	7.36	7.09	8.07	11.06	8.63	6.87	0.96	1.10	1.14	0.78	0.62	0.80
	** *GLN1-1* **	**460.01**	**647.77**	**849.01**	**440.98**	**494.79**	**972.39**	**1.41**	**1.85**	**1.31**	**1.12**	**2.21**	**1.97**
	*GLN1-2*	166.54	216.14	217.36	86.23	122.44	200.58	1.30	1.31	1.01	1.42	2.33	1.64
	*GLN1-3*	1.24	0.10	12.72	27.75	199.31	27.53	0.08	10.26	**127.20**	7.18	0.99	0.14
	*GLN2*	204.01	131.84	232.76	41.54	125.62	163.13	0.65	1.14	1.77	3.02	3.93	1.30
GDH	*GDH1*	31.20	55.95	49.60	20.53	35.14	29.92	1.79	1.59	0.89	1.71	1.46	0.85
	** *GDH2* **	**33.42**	**40.07**	**57.72**	**95.67**	**103.85**	**68.10**	**1.20**	**1.73**	**1.44**	**1.09**	**0.71**	**0.66**
GOGAT	*Os01g0681900*	68.35	32.71	70.93	75.89	116.38	85.39	0.48	1.04	2.17	3.64	5.08	1.40
	*Os05g0555600*	31.51	15.91	21.19	3.42	12.45	17.39	0.50	0.67	1.33	1.53	1.13	0.73
	** *GLSF* **	**316.83**	**253.58**	**207.58**	**110.49**	**159.29**	**169.29**	**0.80**	**0.66**	**0.82**	**1.44**	**1.53**	**1.06**
NR	** *NIA1* **	**15.40**	**17.22**	**18.32**	**21.42**	**39.92**	**18.05**	**1.12**	**1.19**	**1.06**	**1.86**	**0.84**	**0.45**
	*CB5-A*	24.93	26.64	23.79	44.40	32.88	21.37	1.07	0.95	0.89	0.74	0.48	0.65
NirA	** *Os01g0357100* **	**15.07**	**14.91**	**18.20**	**27.99**	**16.81**	**15.52**	**0.99**	**1.21**	**1.22**	**0.60**	**0.55**	**0.92**
CA	** *CA1* **	**138.96**	**79.76**	**128.36**	**27.52**	**74.26**	**67.40**	**0.57**	**0.92**	**1.61**	**2.70**	**2.45**	**0.91**
	*-*	249.36	204.38	150.68	112.35	159.95	139.68	0.82	0.60	0.74	1.42	1.24	0.87
	*-*	4.22	5.02	12.40	0.73	8.37	13.92	1.19	2.94	2.47	11.47	19.07	1.66

## Data Availability

The data presented in this study are available on request from the corresponding author.

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
