# Peer review of "Effects of Biological Nitrogen Metabolism on Glufosinate-Susceptible and -Resistant Goosegrass (Eleusine indica L.)"

_ijms, 2023, doi:10.3390/ijms241813791_

Round 1
Reviewer 1 Report
Dear authors,
I appreciate your efforts and time in preparing the manuscript. However, the manuscript needs to be reorganized prior to acceptance. Please find below my comments and suggestions:
1. Please provide the year and the coordinates of the place of collection.
2. The authors state that “the plants were sprayed at 5-6 leaf stage”. However, in fields, glufosinate treatment is usually done in 3-4 leaf stage.
3. Also, usually weight of aboveground fresh biomass is taken 28 days after treatment, the authors took it after 14 days. Any specific reason?
4. Why Actin was the reference gene used in qRT-PCR? Any relevant reference?
5. Discussion is well-written, but very brief and missed out on many important points such as how this study reduced the research gaps, the discussion of recent findings, and also how the study correlates transcriptome and proteome, etc.
6. Figure 4 is of very low quality.
7. Figure 8 can be shifted to the supplementary or discussion part.
8. Please arrange properly: tables 3 and 4.
9. Language proofreading is required.
Extensive editing of the English language is required.
Author Response
RESPONSE TO REVIEWER 1
of “Effects of biological nitrogen metabolism on glufosinate-susceptible and -resistant goosegrass (Eleusine indica L.)”
by Qiyu Luo, Hao Fu, Fang Hu, Shiguo Li, Qiqi Chen, Shangming Peng, Chunyi Yang, Yaoguang Liu, Yong Chen
25th of August of 2023
We would like to thank you for your kind review of the manuscript. You raise specific issues that are very helpful for improving the manuscript. We agree with all your comments and we have revised our manuscript accordingly.
We responded below in detail to each of the comments including why we designed the research and how we revised things. Most of the language used in this version of the paper has been re-edited with quality. We crafted it to state the discussion of our work more sufficiently than before. Please, find below your comments repeated in italics and our responses inserted after each comment.
Moreover, we are willing to finish the revised version of the manuscript including any further suggestions that the reviewers may have.
Sincerely,
Qiyu Luo and Yong Chen
Response to comments
- Please provide the year and the coordinates of the place of collection.
We agree and add the information about natural goosegrass seeds as follows:
“Natural goosegrass seed samples were collected from the following areas: Ningxi Field (in the town of Ningxi in the district of Zengcheng, Guangzhou, Guangdong, China) (NX, E113º49´N23º08´, originally collected in 2015; purified in South China Agricultural University, E113º36´N23º16´), the town of Shaxi (Chaozhou, Chaoshan Area, Guangdong, China) (CS02, E166º38´N23º41´, originally collected in 2015; purified in South China Agricultural University, E113º36´N23º16´), and the University of Western Australia (Perth, Australia) (AUS, purified in Perth, E115º52'S31º52'; originally collected in the Jerantut farm of Malaysia in 2013, E102º22'N3º56'). ”
- The authors state that “the plants were sprayed at 5-6 leaf stage”. However, in fields, glufosinate treatment is usually done in 3-4 leaf stage.
Yes, glufosinate treatment is usually done in 3-4 leaf stage both in fields and the pot trial. Meanwhile, some pot trials also had options for plant research on glufosinate treatment at the 6- to 8-leaf stage (Carvalho-Moore et al., 2022). The taller the goosegrass, the harder it is to be controlled by glufosinate (Corbett et al., 2004). The taller the goosegrass, the stronger the resistance may detect in this study. We originally wanted to obtain more goosegrass samples in the same batch to measure various physiological and biochemical indexes.
- Also, usually weight of aboveground fresh biomass is taken 28 days after treatment, the authors took it after 14 days. Any specific reason?
Yes, 28 days is a usually used period. We took it after 14 days (2 weeks or half of 28 days). The reason was the effective goosegrass control could be visual after glufosinate application for 14 days. Many plants produced new shoots and panicles in 2 weeks after glufosinate treatment (Wolter et al., 2023). We referred to the processing methods of some literature and additionally marked them in the revision (Burke et al., 2005; Wilson et al., 2007; Wolter et al., 2023).
- Why Actin was the reference gene used in qRT-PCR? Any relevant reference?
Yes, it’s a thoughtful question. Actin is one of reference gene (Huggett et al., 2005). The eIF4A is also considered as an alternative reference gene of plants in different reference (Gutierrez et al., 2008; Chen JC et al., 2017). We chose the Actin amplified from goosegrass as an internal control with the same sequence following our previous reports (Luo et al., 2019). In this revision, we added the relevant references as follows: “Actin-F/R (5’-AACATCGTTCTCAGTGGTGG-3’/5’-CCAGACACTGTACTTCCTTTCA-3’) as the primer of an inner reference gene from the goosegrass (Huggett et al., 2005; Gutierrez et al., 2008; Chen JC et al., 2017; Luo et al., 2019). ”
- Discussion is well-written, but very brief and missed out on many important points such as how this study reduced the research gaps, the discussion of recent findings, and also how the study correlates transcriptome and proteome, etc.
We agree. We need to highlight the value of this study. Therefore, We checked all the references related to the keywords “glufosinate and goosegrass”. There were relatively few references on discoveries of transcriptome and proteome in this field. Then we summarized the findings of these references in the revision as follows:
“We Different mechanisms for the resistance to glufosinate in other glufosinate-resistant species showed that the function of the bar gene was involved with penetration on resistance of a Triticum aestivum line (Rojano-Delgado et al., 2013). And the Italian ryegrass (Lolium perenne ssp. multiflorum) was resistant to glufosinate with one amino acid substitution in GS2 but still poorly understood physiological and genetic mechanisms (Avila-Garcia et al., 2011, 2012; Ghanizadeh et al., 2015; Karn et al., 2018). It was followed by a report of glufosinate-resistant rigid ryegrass (Lolium rigidum Gaud.) in Greece (Travlos et al., 2018). Meanwhile, RNA-Seq transcriptome analysis of Palmer amaranth (Amaranthus palmeri) was carried out to screen candidate genes related to glufosinate tolerance, such as P450 genes (Salas-Perez et al., 2018). The chloroplastic glutamine synthetase enzyme (GS2) of resistant Palmer amaranth responded to glufosinate resistance by enhancing amplification and expressions without mutations (Carvalho-Moore et al., 2022). The activity of protoporphyrinogen oxidase (PPO) inhibitors and the generation of reactive oxygen species (ROS) were related to the action mechanism of glufosinate, including in the species of Palmer amaranth (Takano et al. 2019, 2020a, 2020b, 2020c). Long-term research in the control of goosegrass in glufosinate-resistant cotton found that goosegrass tolerance to glufosinate may be caused by translocation limitation for most of the radioactivity retained in the leaves at about 90% (Everman et al., 2007; 2009a; 2009b; Sharpe et al., 2019). Recently, a novel point mutation was found in the goosegrass mutant EiGS1-1 with a Ser59Gly substitution (Zhang et al. 2022). Taken overall, the effects of biological nitrogen metabolism on glufosinate-susceptible and -resistant goosegrass would play a role in future research for mechanisms of glufosinate-resistance in terms of the metabolic aspects.”
- Figure 4 is of very low quality.
We agree and we revise it. We also checked the quality of other figures.
- Figure 8 can be shifted to the supplementary or discussion part.
We agree. The Figure 8 with the corresponding text was shifted to the discussion part as follows: “A graphic model of the differentially expressed genes and proteins in the nitrogen metabolism pathways under glufosinate stress in susceptible and resistant goosegrasses (NX, CS, and AUS) was analyzed by KEGG enrichment (Figure 8). The model visualizes the changes in differential expression in the nitrogen metabolism pathways in the same way as their original pathways (Figure S2). Generally, the transcription levels of all six genes (GS, GOGAT, GDH, NirA, NR, and CA) were affected in susceptible goosegrasses (T-NX vs. CK-NX) with GDH upregulated, including their protein levels of GDH, NR, and CA. However, glufosinate stress played a minor role in the nitrogen metabolism of resistant goosegrasses (T-AUS vs. CK-AUS), with only the transcription levels of GS being upregulated. This suggests that the glufosinate-resistant mechanism patterns in goosegrass still ultimately depend on changes in GS, which implies the importance of conducting further research on GS in glufosinate-resistant goosegrass (AUS) compared to susceptible goosegrass (NX). ”
- Please arrange properly: tables 3 and 4.
Yes, we agree and revise it. The tables have been standardized with the format to display numerical values and visual results.
- Language proofreading is required.Comments on the Quality of English Language Extensive editing of the English language is required.
We agree and try our best to rewrite it. And we are grateful for receiving the help of English language editing by MDPI.

Reviewer 2 Report
This manuscript aims to explain the mechanism of E. indica resistance to glufosinate herbicide based on two resistant biotypes and one susceptible.
Comments:
Authors mention that E. indica is one of the most troublesome weeds in the world. However, this weed is typical for warmer bioregions, e.g., subtropical. That is why a better botanical and biological characteristic of the species would be welcome in the Introduction part.
There were two different biotypes of resistant E. indica studied. It would be nice to summarize the main differences between them in the table, which would show what are the similarities and differences in the resistance mechanism.
Please explain the abbreviations used in the Figures and abstract.
Figure 4 is of low quality.
Discussion - please add that AUS and CS are two biotypes of EW. indica resistant to glufosinate.
The discussion is poor in content. Adding literature sources to support your findings regarding the different mechanisms for the resistance to glufosinate would be nice, a. i., reference to other glufosinate-resistant species.
Also, the conclusions are too concise. But I understand that this is because of a very short Discussion.
Author Response
RESPONSE TO REVIEWER 2
of “Effects of biological nitrogen metabolism on glufosinate-susceptible and -resistant goosegrass (Eleusine indica L.)”
by Qiyu Luo, Hao Fu, Fang Hu, Shiguo Li, Qiqi Chen, Shangming Peng, Chunyi Yang, Yaoguang Liu, Yong Chen
25th of August of 2023
We would like to thank you for your kind review of the manuscript. Your inputs are very helpful for improving the manuscript. We agree with all your comments and we have revised our manuscript accordingly.
We respond to each of the comments in detail below. We are already crafting a revised version of the paper that added more information about our work in the introduction and discussion. Moreover, we included all your suggestions and clarified the text. Please, find below your comments repeated in italics and our responses inserted after each comment.
We hope that you will find our responses to your comments satisfactory.
Sincerely,
Qiyu Luo and Yong Chen
Response to comments
- A better botanical and biological characteristic of the species would be welcome in the Introduction part.
We agree and we add the information of botanical and biological characteristics as follows: “Goosegrass (Eleusine indica L. Gaertn.) is one of the worst weeds (a self-pollinating monoecious species) in the world, which can produce large quantities of seeds germinating at any time of the year and being carried in clusters (Kurniadie et al., 2023). The taller the goosegrass, the harder it is to be controlled by glufosinate (Corbett et al., 2004). It was first found resistant to glufosinate in Malaysia in 2009 (Jalaludin et al., 2010, 2014).”
- There were two different biotypes of resistant E. indica studied. It would be nice to summarize the main differences betweenthem in the table, which would show what are the similarities and differences in the resistance mechanism.
We agree, and we mark them in the table with bold values. We add the information on similarities and differences as follows: “The tables showed differences and similarities between two different biotypes of goosegrass (AUS and CS). On the one hand, the expression levels of genes and proteins in the GS family exhibit greater differences, such as 127.20-fold GLN1-3 (1.75-fold GLN1-3) in CK-AUS/CK-CS and 1.97-fold GLN1-1 (1.26-fold GLN1-1) in T-AUS/T-CS. On the other hand, it remained consistent that the expression levels of genes and proteins in the GS family had the highest expression level in the five families, especially compared to the NirA family. The highest expression levels were found in the five genes (Bold Mark), with higher expression levels than other genes of each family in both AUS and CS goosegrasses.”
- Please explain the abbreviations used in the Figures and abstract.
We agree and add the information of abbreviations as follows: GS, glutamine synthetase; GOGAT, glutamate 2-oxoglutarate aminotransferase; GDH, glutamate dehydrogenase in figures and abstract.
- Figure 4 is of low quality.
We agree and we revise it. We also checked the quality of other figures.
- Discussion - please add that AUS and CS are two biotypes of EW. indica resistant to glufosinate.
We agree and add the information as the top sentence of the second paragraph in the discussion part as follows: “AUS and CS are two biotypes of Eleusine indica L. resistant to glufosinate. During goosegrass tolerance to glufosinate stress, AUS and CS employ distinct mechanisms of molecular regulation in the nitrogen metabolism.”
- Adding literature sources to support your findings regarding the different mechanisms for the resistance to glufosinate would be nice, a. i., reference to other glufosinate-resistant species.
Yes, we agree and try our best to add more information as follows:
“We Different mechanisms for the resistance to glufosinate in other glufosinate-resistant species showed that the function of the bar gene was involved with penetration on resistance of a Triticum aestivum line (Rojano-Delgado et al., 2013). And the Italian ryegrass (Lolium perenne ssp. multiflorum) was resistant to glufosinate with one amino acid substitution in GS2 but still poorly understood physiological and genetic mechanisms (Avila-Garcia et al., 2011, 2012; Ghanizadeh et al., 2015; Karn et al., 2018). It was followed by a report of glufosinate-resistant rigid ryegrass (Lolium rigidum Gaud.) in Greece (Travlos et al., 2018). Meanwhile, RNA-Seq transcriptome analysis of Palmer amaranth (Amaranthus palmeri) was carried out to screen candidate genes related to glufosinate tolerance, such as P450 genes (Salas-Perez et al., 2018). The chloroplastic glutamine synthetase enzyme (GS2) of resistant Palmer amaranth responded to glufosinate resistance by enhancing amplification and expressions without mutations (Carvalho-Moore et al., 2022). The activity of protoporphyrinogen oxidase (PPO) inhibitors and the generation of reactive oxygen species (ROS) were related to the action mechanism of glufosinate, including in the species of Palmer amaranth (Takano et al. 2019, 2020a, 2020b, 2020c). Long-term research in the control of goosegrass in glufosinate-resistant cotton found that goosegrass tolerance to glufosinate may be caused by translocation limitation for most of the radioactivity retained in the leaves at about 90% (Everman et al., 2007; 2009a; 2009b; Sharpe et al., 2019). Recently, a novel point mutation was found in the goosegrass mutant EiGS1-1 with a Ser59Gly substitution (Zhang et al. 2022). Taken overall, the effects of biological nitrogen metabolism on glufosinate-susceptible and -resistant goosegrass would play a role in future research for mechanisms of glufosinate-resistance in terms of the metabolic aspects.”
- Also, the conclusions are too concise. But I understand that this is because of a very short Discussion.
We agree and improve it as follows: “ In conclusion, this study clarified the nitrogen metabolism pathway that influences the glufosinate-resistant goosegrass in response to stress. The putative target genes GLN1-1, GDH2, GLSF, NIA1, Os01g0357100, and CA1 were screened out. The expression patterns of the putative genes involved in glufosinate-resistant goosegrass could serve as a basis for future functional verification in transgenic plants.”

Round 2
Reviewer 1 Report
Dear authors, I appreciate the revisions done. Good job. There are some points, such as, in some places, biotypes are written as "bio-types". Also, there are a few other typo errors. Please check them, prior to submitting the final version. I now recommend the manuscript for publication. Congratulations...!!
Author Response
Dear Reviewer 1,
We would like to thank you for your kind review and encouragement. We tried our best efforts to check all “bio-types” and revised all typo errors. We also numbered all references in order of appearance in the manuscript.
Thank you again for the time and help you have invested in this round of the review report.
Best regards,
Qiyu Luo and Yong Chen